# Machine Learning Based on MRI DWI Radiomics Features for Prognostic Prediction in Nasopharyngeal Carcinoma

**DOI:** 10.3390/cancers14133201

**Published:** 2022-06-30

**Authors:** Qiyi Hu, Guojie Wang, Xiaoyi Song, Jingjing Wan, Man Li, Fan Zhang, Qingling Chen, Xiaoling Cao, Shaolin Li, Ying Wang

**Affiliations:** 1Department of Nuclear Medicine, The Fifth Affiliated Hospital of Sun Yat-sen University, Zhuhai 519099, China; huqy28@mail2.sysu.edu.cn (Q.H.); polaris0817@163.com (J.W.); chenqling23@mail2.sysu.edu.cn (Q.C.); caoxling@mail2.sysu.edu.cn (X.C.); 2Department of Radiology, The Fifth Affiliated Hospital of Sun Yat-sen University, Zhuhai 519099, China; wanggj5@mail.sysu.edu.cn; 3Guangdong Provincial Key Laboratory of Biomedical Imaging, The Fifth Affiliated Hospital of Sun Yat-sen University, Zhuhai 519099, China; songxy29@mail2.sysu.edu.cn (X.S.); liman26@mail.sysu.edu.cn (M.L.); 4Department of Head and Neck Oncology, The Cancer Center of the Fifth Affiliated Hospital of Sun Yat-sen University, Zhuhai 519099, China; zhangfan26@mail.sysu.edu.cn

**Keywords:** radiomics, nasopharyngeal carcinoma, diffusion-weighted imaging, prognostic prediction, heterogeneity

## Abstract

**Simple Summary:**

In the past, radiomics studies of nasopharyngeal carcinoma (NPC) were only based on basic MR sequences. Previous studies have shown that radiomics methods based on T2-weighted imaging and contrast-enhanced T1-weighted imaging have been successfully used to improve the prognosis of patients with nasopharyngeal carcinoma. The purpose of this study was to explore the predictive efficacy of radiomics analyses based on readout-segmented echo-planar diffusion-weighted imaging (RESOLVE-DWI) which quantitatively reflects the diffusion motion of water molecules for prognosis evaluation in nasopharyngeal carcinoma. Several prognostic radiomics models were established by using diffusion-weighted imaging, apparent diffusion coefficient maps, T2-weighted and contrast-enhanced T1-weighted imaging to predict the risk of recurrence or metastasis of nasopharyngeal carcinoma, and the predictive effects of different models were compared. The results show that the model based on MRI DWI can successfully predict the prognosis of patients with nasopharyngeal carcinoma and has higher predictive efficiency than the model based on the conventional sequence, which suggests MRI DWI-radiomics can provide a useful and alternative approach for survival estimation.

**Abstract:**

Purpose: This study aimed to explore the predictive efficacy of radiomics analyses based on readout-segmented echo-planar diffusion-weighted imaging (RESOLVE-DWI) for prognosis evaluation in nasopharyngeal carcinoma in order to provide further information for clinical decision making and intervention. Methods: A total of 154 patients with untreated NPC confirmed by pathological examination were enrolled, and the pretreatment magnetic resonance image (MRI)—including diffusion-weighted imaging (DWI), apparent diffusion coefficient (ADC) maps, T2-weighted imaging (T2WI), and contrast-enhanced T1-weighted imaging (CE-T1WI)—was collected. The Random Forest (RF) algorithm selected radiomics features and established the machine-learning models. Five models, namely model 1 (DWI + ADC), model 2 (T2WI + CE-T1WI), model 3 (DWI + ADC + T2WI), model 4 (DWI + ADC + CE-T1WI), and model 5 (DWI + ADC + T2WI + CE-T1WI), were constructed. The average area under the curve (AUC) of the validation set was determined in order to compare the predictive efficacy for prognosis evaluation. Results: After adjusting the parameters, the RF machine learning models based on extracted imaging features from different sequence combinations were obtained. The invalidation sets of model 1 (DWI + ADC) yielded the highest average AUC of 0.80 (95% CI: 0.79–0.81). The average AUCs of the model 2, 3, 4, and 5 invalidation sets were 0.72 (95% CI: 0.71–0.74), 0.66 (95% CI: 0.64–0.68), 0.74 (95% CI: 0.73–0.75), and 0.75 (95% CI: 0.74–0.76), respectively. Conclusion: A radiomics model derived from the MRI DWI of patients with nasopharyngeal carcinoma was generated in order to evaluate the risk of recurrence and metastasis. The model based on MRI DWI can provide an alternative approach for survival estimation, and can reveal more information for clinical decision-making and intervention.

## 1. Introduction

Nasopharyngeal carcinoma (NPC) is an epithelial malignancy with distinctive geographic distribution [1]. Over 130,000 patients were newly diagnosed with NPC in 2020, among which more than 70% were located in East and South East Asia [1,2]. Even with advancements in screening and treatments, approximately 5–15% of patients exhibit local recurrence, and 15–30% of NPC patients experience metastatic spread after standard treatment [3]. Therefore, identifying the reliable predictive factors associated with prognosis is necessary. In the last few decades, tumor heterogeneity has continued to be a crucial factor influencing prognosis [4]. At present, the clinical formulation of treatment primarily depends on the TNM staging system. However, similar clinical treatment can result in distinct clinical outcomes for NPC patients with the same TNM stage [5], indicating that the system merely reflects the anatomic invasion and fails to adequately unmask tumor heterogeneity.

Moreover, some specific blood metabolites or cellular and genetic parameters are used to predict the prognosis of nasopharyngeal carcinoma patients, such as EBV-DNA, LDH, ALP, HOPX, miRNAs, and gene expression, etc. [6,7,8,9,10]. Importantly, EBV-DNA and several pretreatment inflammatory biomarkers have been considered as independent prognostic factors for patients with NPC, including lymphocyte and neutrophil counts, and the neutrophil-to-lymphocyte ratio (NLR), etc. [11]. Nevertheless, the former biomarkers present instability and non-specificity, whereas the routine application of the latter parameter modality is restricted by the expensive cost. Therefore, a low-cost, convenient, and accurate approach that can evaluate heterogeneity and prognosis is urgently needed.

The radiomics technique has emerged as a promising approach to the conversion of images into high-dimensional and quantitative features [12]. Radiomics analysis based on clinical images can provide additional information about tumor heterogeneity steadily and accurately, and can thus offer clinical support for decision making, thereby improving tumor treatment with an economic and non-invasive approach [13]. The radiomics model based on MRI to predict the prognosis of patients with nasopharyngeal carcinoma has been observed, and has exported great value in risk stratification and prognosis evaluation [14,15,16]. However, related studies only extract image features from basic MRI sequences. As a functional imaging technique, DWI can quantitatively demonstrate the diffusion motion of water molecules in the tissue microenvironment, and can detect tissue cellularity, microstructures, and microvasculature at the sub-voxel level, thereby revealing additional internal features of the tumor in order to uncover vital prognostic information [17]. It has been frequently used in clinical trials to report on differential diagnosis, staging, therapeutic evaluation, and prognostic prediction in oncology [18].

In the past, DWI images suffered from insufficient image quality, including obvious artifacts, limited resolution, and blurred images, which may hinder their routine application in radiomics in the head and neck [19]. However, readout-segmented imaging (RS-EPI) approaches have now been introduced to perform high-resolution diffusion-weighted MRI (HR-DWI), and have greatly improved image quality with a higher resolution and fewer artifacts than the extensively adopted single-shot imaging (SS-EPI) DWI [20]. This improvement is achieved by shortening the data-acquisition time and dividing the k-space into multiple interleaved acquisitions in order to diminish the accumulation of phase errors in the phase-encoding direction. Previous studies have shown that a radiomics model based on DWI MRI can accurately reveal the individual prognosis in several cancers, such as bladder, hepatocellular, and prostate cancers [21,22,23].

According to the literature searched, whether radiomics based on a DWI sequence can extract the tumor heterogeneity of nasopharyngeal carcinoma and evaluate the risk of recurrence and metastasis remains uncertain. Accordingly, we performed the present study to visualize the heterogeneity and disclose the prognosis of nasopharyngeal carcinoma through radiomics analyses based on the RESOLVE-DWI sequence. Furthermore, we sought to compare and combine the radiomics model based on the RESOLVE-DWI sequence and conventional sequence (T2WI and CE-T1WI) in order to provide more clinical decision-making and intervention information.

## 2. Materials and Methods 

### 2.1. Patients

Approval for this retrospective study was obtained from the Ethics Review Committee of the Fifth Affiliated Hospital of Sun Yat-sen University (ClinicalTrials.gov Identifier: NCT05112510). The Committee exempted the informed consent concurrently. A total of 154 patients with untreated NPC confirmed by pathological examination between March 2014 and June 2018 were enrolled, including 15 patients with local or regional tumor recurrence and 28 patients with distant metastasis (1 of the patients had local recurrence and metastases simultaneously).

The collected clinical features included age, gender, tumor size (T), nodal status (N), metastases (M), TNM staging, and histological subtypes. The staging was based on the Eighth American Joint Committee on Cancer TNM staging manual [24]. According to the criteria from the World Health Organization (WHO), the histological subtypes were classified into three patterns: keratinizing squamous cell carcinoma (type I), nonkeratinizing differentiated carcinoma (type II), and nonkeratinizing undifferentiated carcinoma (type III) [25].

### 2.2. Inclusion and Exclusion Criteria

The eligibility criteria for patient enrollment were as follows: (1) patients with NPC confirmed by pathological examination; (2) patients with complete MR images and clinical data; (3) patients who did not receive chemotherapy, radiotherapy, or surgery before their MRI scans. Patients were removed by applying the following exclusion criteria: (1) the periodical follow-up data were incomplete; (2) poor image quality; and (3) patients with a concomitant or previous history of cancer.

### 2.3. Endpoints

Failure-free survival (FFS) was defined as the primary endpoint in this study, and it was considered from the first date of the MR scan, and ended with the progression. Local recurrence was diagnosed through pathological examinations. If any medical report indicated distant metastasis, the suspected site of involvement was subjected to extra histological confirmation. In the case of failed biopsy or no biopsy, regular follow-up was attempted. Distant metastasis was diagnosed when the enlargement of the lesions was observed.

### 2.4. MRI Acquisition

All 154 patients underwent a series of MRI scans. The sequences included axial T2-weighted imaging (T2WI), contrast-enhanced T1-weighted imaging (CE-T1WI), axial DWI (*b* = 800 s/mm^2^), and ADC mapping. The MRI scanning was performed on a Magnetom Trio 3.0T MRI scanner (Siemens Medical, Erlangen, Germany). An eight-channel head and neck coil was adopted in order to collect the signals. The scanning range was from the skull base to the subclavian region. The conventional MRI sequence included axial T2WI and CE-T1WI. The contrast agent was a Gadobutrol injection. 

The following parameters were set for the axial T2WI: TR/TE, 3760 ms/72 ms; field of view (FOV), 230; matrix size, 320 × 224; layer thickness, 5 mm; interlayer spacing, 1 mm; bandwidth, 340 Hz; acquisition time, 3 min and 23 s; number of excitations (NEX), 2; and resolution, 0.7 × 0.7.

The following parameters were set for CE-T1WI: TR/TE, 4660 ms/10 ms; FOV, 230; matrix size, 320 × 224; layer thickness, 5 mm; interlayer spacing, 1 mm; bandwidth, 347 Hz; acquisition time, 2 min 49 s; NEX, 3; and resolution, 0.7 × 0.7.

The following parameters were set for RESOLVE-DWI: RS-EPI, TR/TE, 3800 ms/65 ms; matrix size, 192 × 192; layer thickness, 4 mm; interlayer spacing, 0.6 mm; bandwidth, 521 Hz; acquisition time, 2 min 55 s; segmented readout times, 9; and *b* = 0, 800 s/mm^2^. The ADC maps were automatically generated from the MRI system. 

### 2.5. Segmentation and Feature Extraction

All of the regions of interest (ROIs) of the images were manually segmented in all of the slices by two radiologists: one with 5 years of clinical experience and the other with 15 years. A total of 5636 features were extracted. Manual segmentation and relative feature extraction were both conducted in the Radcloud platform (https://mics.radcloud.cn, accessed: 23 May 2022). The intraclass correlation coefficient (ICC) in 20 patients was calculated in order to assess the intra- and inter-observer variability for consistency. Features with an ICC below 0.75 were excluded.

### 2.6. Radiomics Feature and Model Selection

All of the feature columns with the same numerical values were eliminated, and normalization processing at the order of magnitude was performed on all of the features. The extracted features were screened by Random Forest (RF), which creates a decision tree such that the suboptimal segmentation is performed by introducing randomness; this has been adopted extensively in radiomics based on its excellent performance in classification tasks [26]. The workflow for feature selection by Random Forest can be summarized as follows. First, the differential clinical characteristics were added and set as dummy variables. The top 100 features were screened according to importance. Then, the top 10 features in terms of improving the model’s predictive power were retained after the cyclical inclusion of each feature with a forward stepwise approach by the RF method. Finally, the features of each model were limited to 10. The training set was randomly split with the k-fold cross-validation method: the training set was divided into five subsets, and one of the K-fold sample sizes was *N* = 26 (two-folds: *N* = 27).

The differences in clinical factors between the two groups were investigated by one-way analysis of variance in SPSS (version 25.0, IBM Corp, Armonk, NY, USA). The Chi-square test was used for categorical variables, and the Mann–Whitney U test was used for continuous variables. Hierarchical variables used the Wilcoxon symbol order and test. Python software was performed to screen, choose, and build the machine learning models based on the screened features.

Five of the existing mainstream algorithms (Logistic Regression, kNN, Naive Bayes, Random Forest, and XGB Classifier) were chosen for training and validation. In order to obtain a more robust model, we applied five-fold cross-validation to calculate the average AUC of the training sets and the average AUC of the validation sets. The obtained results were presented as the average AUC of cross-training set and the average AUC of the cross-validation set. The major parameters of the corresponding models were adjusted using GridSearchCV. The model was chosen according to the average AUC of the cross-validation set [27].

### 2.7. Prediction Model Building

The selected models mentioned above were trained and validated based on the screened features from different sequence combinations, and the parameters were adjusted. All of the major parameters, such as criterion, max_depth, min_samples_leaf, min_samples_split, max_features, and min_impurity decrease, were adjusted within a large range. The OOB_score was chosen as the evaluation criterion, resulting in the parameters of all of the final models. 

Models after the parameter adjustment were used for five-fold cross-validation, and were compared in order to obtain the optimal AUC of different sequence combinations. Accordingly, the optimal machine learning models based on the extracted imaging features from different sequence combinations were built, including model 1 (DWI + ADC), model 2 (T2WI + CE-T1WI), model 3 (DWI + ADC + T2WI), model 4 (DWI + ADC + CE-T1WI), and model 5 (DWI + ADC + T2WI + CE-T1WI). The study workflow is briefly displayed in Figure 1.

## 3. Results

### 3.1. Clinical Characteristics Analysis

In the present study, 154 patients were included, including 43 females (29%) and 111 males (71%), with a median age of 47 years (19–68). The most common histopathological subtype refers to undifferentiated nonkeratinizing carcinoma (SCC, 80.6%). The relapsed or metastatic group and the non-relapsed or metastatic group presented significant differences in the N, M, and TNM stages (*p* < 0.05). The patient characteristics are presented in Table 1.

### 3.2. Machine Learning Model Selection

Five-fold cross-validation was carried out using Logistic Regression, kNN, Naive Bayes, Random Forest, and XGB Classifier, and the results show that the AUC obtained using the RF method is the highest among the different sequence combinations. The results are shown in Figure 2. Therefore, the RF machine learning model was chosen to compare the predictive performances of the different sequence combination models.

### 3.3. Prediction Performance of the Models

Concerning the construction and results of different sequence-combination models, the N and M stages were added according to the dissimilarity tests of the clinical variables, and they were set as dummy variables. The top 100 features were screened by the importance of the RF method. Then, the top 10 features in terms of improving the model’s predictive power were retained after the cyclical inclusion of each feature with a forward stepwise approach. The selected features and importances are shown in Figure 3. The selected features were used to construct the RF machine learning prediction model. In order to obtain a more robust outcome, we applied five-fold cross-validation, and the AUC of the validation set in the machine learning model was obtained based on different sequence combinations using the RF method.

In order to obtain a more robust outcome, we applied five-fold cross-validation to train and validate the RF machine learning model. After adjusting the parameters, the average AUC of the validation set in the RF machine learning model was obtained based on the extracted imaging features from different sequence combinations. The mean AUCs of the five-fold cross-validation sets of model 1 (DWI + ADC), model 2 (T2WI + CE-T1WI), model 3 (DWI + ADC + T2WI), model 4 (DWI + ADC + CE-T1WI), and model 5 (DWI + ADC + T2WI + CE-T1WI) were 0.80 (95% CI: 0.79–0.81), 0.72 (95% CI: 0.71–0.74), 0.66 (95% CI: 0.64–0.68), 0.74(95% CI: 0.73–0.75), and 0.75 (95% CI: 0.74–0.76), respectively. The average AUC of each model in validation set is shown in Figure 4. The performances of the radiomics models in the validation set are shown in Table 2.

Based on the results, the RF model based on the extracted features from the DWI and ADC images has higher prognostic prediction efficacy than the RF model based on T2WI and T1WI images. Moreover, the RF model based on the extracted features from DWI, ADC, and T2WI presents better predictive performance for prognosis than the RF model based on DWI, ADC, and CE-T1WI. Finally, the results indicated that the RF model based on the extracted features from the multiple-sequence combination of DWI, ADC, T2WI, and CE-T1WI did not display optimal effects in the prediction of the recurrence and metastasis of nasopharyngeal carcinoma.

## 4. Discussion

Radiomics models based on MRI features in nasopharyngeal carcinoma (NPC) can predict the prognosis and therapeutic responses [28], but these models were constructed based on basic MR sequences (e.g., T2WI, T1WI, and CE-T1WI). Studies with a radiomics approach based on DWI images in nasopharyngeal carcinoma remain to be explored. Considering that the foregoing radiomics research focuses on tumor heterogeneity and the prognosis of NPC mainly based on T2WI and CE-T1WI [15,16,29,30,31,32], we attempted to compare and combine the radiomics model based on RESOLVE-DWI simultaneously with T2WI and CE-T1WI. This process aims to determine the optimal machine learning model for the prognostic prediction of NPC.

Extracted features in various MR sequence combinations were adopted in order to predict the recurrence and metastasis risks of NPC patients in the present study. The results show that the average cross-validated AUC of the RF model based on radiomics features extracted from DWI and ADC sequences reached 0.80, and the AUC of RF models based on conventional MR sequences was 0.72. The AUC of model 2 (T2WI + CE-T1WI) of this study in the validation set closely resembled that of Kim et al.’s study [16], which suggests that the AUC of the radiomics model combining T2WI and CE-T1WI sequences was 0.71 for the prediction of progression-free survival in patients with NPC. At the same time, no data from previous studies were comparable to the results of the radiomics model based on the DWI sequence of the present study on account of the rare usage of DWI in radiomics. However, the radiomics features extracted from the DWI and ADC sequences have higher prediction efficacy in terms of the recurrence and metastasis risks of patients. This finding was potentially obtained due to the quantitative features of models extracted from the image, and the DWI can provide more sub-voxel image information about tumor heterogeneity, which reflects the limited Brownian motion and microarchitecture in tumors [17,33]. Moreover, the machine learning model based on features extracted from the DWI, ADC, and CE-T1WI sequences presents a higher forecast performance than the models based on DWI, ADC, and T2WI sequences. This finding was potentially due to CE-T1WI sequences being able to reflect the blood supply and angiogenesis of tumors [34], and to unmask the proliferation state of tumors better than T2WI, making CE-T1WI sequences more relevant for tumor heterogeneity. Finally, we combined DWI, ADC, T2WI, and CE-T1WI sequences in NPC and extracted the relative features from this combination in order to establish RF machine learning models. The average cross-validated AUC of this model was 0.73 for the prediction of the prognosis of NPC, and this value is not higher than that of the RF model based on DWI and ADC sequences. This finding can be attributed to the increase in mixing factors with the increase in sequences.

Notably, high-resolution DWI was applied to extract related features and build the machine learning model for the prediction of the recurrence and metastasis of nasopharyngeal carcinoma. DWI is a proven non-contrast imaging technology that has become a mature quantitative measurement approach for the identification of benign and malignant lesions in routine clinical work [19,35,36]. In malignant tumors, the diffusion of water molecules is often restricted or limited by the high cell density, which exhibited high signals on DWI and a low value on ADC maps. DWI technology can provide quantitative interpretations as well as qualitative interpretations, thereby increasing the specificity of disease diagnosis [17]. The application in radiomics of a single-shot (SS) EPI-DWI technology extensively used to collect DWI images is easily restricted by magnetic susceptibility artifacts, chemical displacement and geometric distortion, limited spatial resolution, and relatively thick sections, especially in head and neck tumors, such as nasopharyngeal carcinoma with artifacts of the skull base [19]. With the improvement in readout-segmented imaging (RS-EPI) technologies, high-resolution DWI (HR-DWI) was applied to clinical work. It remarkably improved the abovementioned problems by using the same diffusion preparation as SS EPI but dividing the K space into several segments in the phase-encoding direction in order to decrease the echo time [20]. Therefore, readout-segmented imaging (RS-EPI) has obvious advantages and is irreplaceable for the diagnosis of tumors at the head and neck compared with (SS) EPI-DWI [19], and the machine learning model based on DWI collected by RS-EPI is more reliable and robust, providing a good foundation to promote its clinical applications.

Additionally, the acquisition of HR-DWI does not require a contrast agent, making it safer than the CE-T1W in daily clinical work. In present practical applications, it has realized technological advantages of increased speed and decreased artifacts, supporting its extensive use in clinical practice. Based on the above analysis, the radiomics method based on RESOLVE-DWI has higher prediction efficacy than the conventional MR sequence regarding the recurrence and metastasis of NPC. The applications of high-resolution DWI in radiomics might be complementary to—and might even replace—the currently used sequences (T2WI, T1WI, and CE-T1WI) in order to provide more high-specificity information and support for clinical decisions.

The present study has some limitations. First, this study involved a few cases, and it was carried out in one hospital. Therefore, a prospective study should be carried out to support the conclusions. Moreover, minor details are hard to depict, which might influence the extraction of features. Finally, the relationship between radiomics features and prognostic outcomes was not explored further in the present study. Relevant data were collected, and the next step for our research is to discover this relationship and further perform survival analysis according to the radiomics model based on DWI sequences.

## 5. Conclusions

The results confirmed that the machine learning model based on features extracted by RESOLVE-DWI and corresponding ADC maps could be used as a prognosis detection tool. These features can help to quantify the heterogeneity of patients with NPC and evaluate the risk of recurrence and metastasis in order to quickly provide supporting evidence and thus aid in making a sound clinical decision in clinical practice.

## Figures and Tables

**Figure 1 cancers-14-03201-f001:**
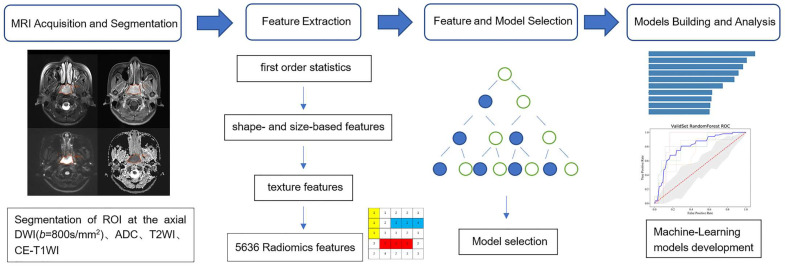
Workflows: (1) MRI acquisition and segmentation; (2) quantitative feature extraction; (3) radiomic feature and model selection; (4) prediction models built based on the extracted imaging features from different sequence combinations.

**Figure 2 cancers-14-03201-f002:**
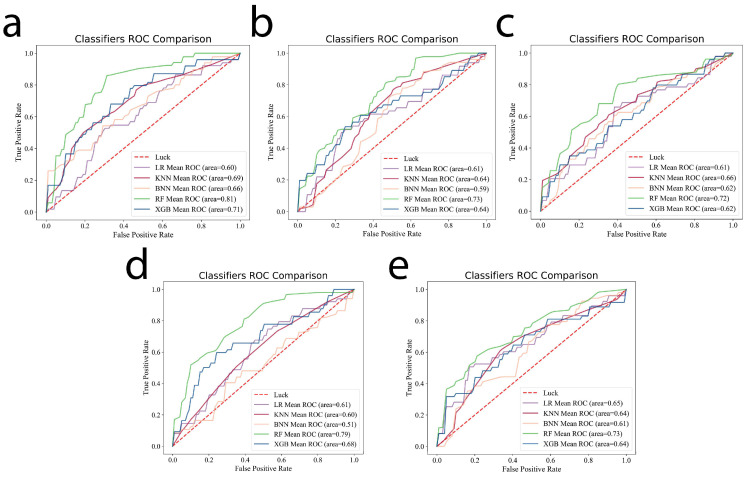
Five existing mainstream algorithms (Logistic Regression, kNN, Naive Bayes, Random Forest, and XGB Classifier) were chosen for the training and validation, which showed that AUC values obtained using the RF method are the highest among all of the models of different sequence combinations: (**a**) DWI + ADC; (**b**) T2WI + CE-T1WI; (**c**) DWI + ADC + T2WI; (**d**) DWI + ADC + CE-T1WI; (**e**) DWI + ADC + T2WI + CE-T1WI.

**Figure 3 cancers-14-03201-f003:**
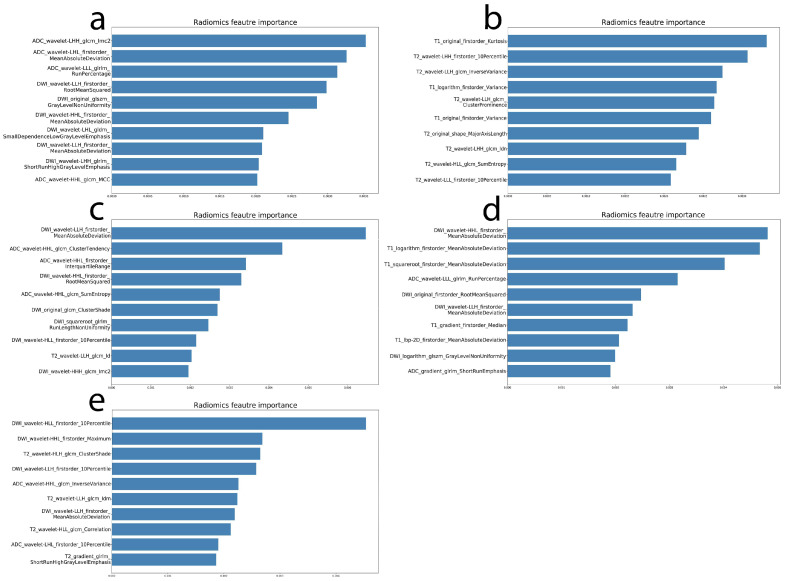
The importance of selected features derived from different sequence combinations: (**a**) DWI + ADC; (**b**) T2WI + CE-T1WI; (**c**) DWI + ADC + T2WI; (**d**) DWI + ADC + CE-T1WI; (**e**) DWI + ADC + T2WI + CE-T1WI.

**Figure 4 cancers-14-03201-f004:**
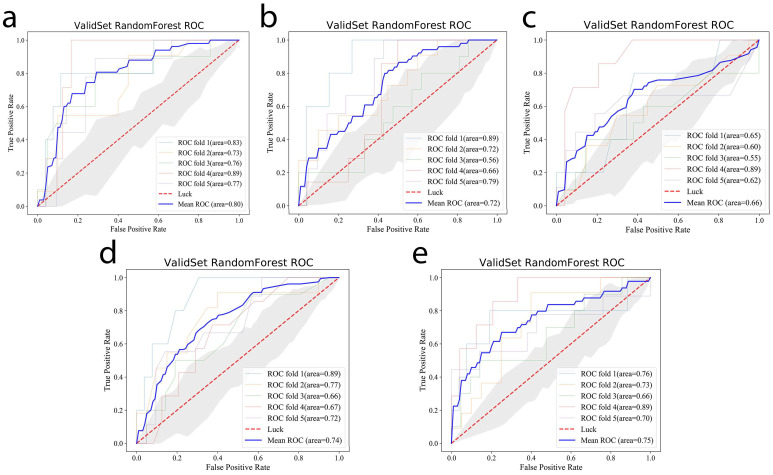
Average AUC values in the validation set of the RF machine learning model based on selected features of model 1 (**a**), model 2 (**b**), model 3 (**c**), model 4 (**d**), and model 5 (**e**).

**Table 1 cancers-14-03201-t001:** Clinical characteristics of the patients with NPC in the relapsed or metastatic group and the non-relapsed or metastatic group.

Characteristics	Type	Positive (%) N = 42	Negative (%) N = 112	*p*-Value
Gender	Male	34	77	0.516
	Female	8	35	
Age (years)	Range	19–68	23–63	0.810
Overall stage	I	0	2	0.026
	II	3	20	
	III	17	56	
	IVa	17	34	
	IVb	5	0	
T stage	I	2	25	0.915
	II	12	22	
	III	13	37	
	IV	15	28	
N stage	0	1	9	0.034
	1	11	48	
	2	21	45	
	3	9	10	
M stage	0	42	107	0.085
	1	0	5	
Histology	WHO type I	0	1	
	WHO type II–III	42	111	0.540

**Table 2 cancers-14-03201-t002:** The performance metrics for five models in the validation set.

Models	AUC	Accuracy	Specificity	Precision
DWI + ADC	0.80 (95% CI: 0.79–0.81)	0.766	0.926	0.620
T2WI + CE-T1WI	0.72 (95% CI: 0.71–0.74)	0.752	0.930	0.520
DWI + ADC + T2WI	0.66 (95% CI: 0.64–0.68)	0.779	0.925	0.689
DWI + ADC + CE-T1WI	0.74(95% CI: 0.73–0.76)	0.766	0.918	0.548
DWI + ADC + T2WI + CE-T1WI	0.75 (95% CI: 0.74–0.76)	0.766	0.923	0.811

## Data Availability

Not applicable.

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
