# Peer review of "Machine Learning Based on MRI DWI Radiomics Features for Prognostic Prediction in Nasopharyngeal Carcinoma"

_cancers, 2022, doi:10.3390/cancers14133201_

Round 1

Reviewer 1 Report

Dear Authors,

The study is well conducted. The machine learning represent an important tool finding different application in medical field. In the last years different paper analyzing the role of machine learning and radiomics in head and neck malignancies.

I suggest to better clarify some passagges:

In the Method section clarify how is working Random Forest.

In the discussion could be interesting if the Authors reports some recents applications of machine learning and radiomicsin head and neck, comparing with their results.

A moderate english revision by a native speaker is mandatory.

Author Response

We responded to each comment and listed the corresponding responses in the attachment.

Reviewer 2 Report

Strength of the paper: 

Artificial intelligence applied to medicine today represents a topic of great interest. This is a very original work on the ability to predict relapses / secondary diseases in nasopharyngeal carcinomas. The value of the proposed study lies in investigating the correlation between specific radiomic features based on diffusion sequences (MRI DWI) and the prognosis of Nasopharyngeal cancer patients. In general, the work is well written and the methodology used is correct.Weakness of the paper:

-  In the introduction some prognostic indices studied in the literature are mentioned. No mention is made of the increasingly important role that peripheral inflammatory biomarkers play in the prognosis of nasopharyngeal carcinomas. A brief reference would ensure greater scientific visibility.

-        Cupp MA, Cariolou M, Tzoulaki I, Aune D, Evangelou E, Berlanga-Taylor AJ. Neutrophil to lymphocyte ratio and cancer prognosis: an umbrella review of systematic reviews and meta-analyses of observational studies. BMC Med. 2020 Nov 20;18(1):360. doi: 10.1186/s12916-020-01817-1. PMID: 33213430; PMCID: PMC7678319.

-        Su L, Zhang M, Zhang W, Cai C, Hong J. Pretreatment hematologic markers as prognostic factors in patients with nasopharyngeal carcinoma: A systematic review and meta-analysis. Medicine (Baltimore). 2017 Mar;96(11):e6364. doi: 10.1097/MD.0000000000006364. PMID: 28296774; PMCID: PMC5369929.

-       The proposed model is useful for evaluating the prognosis of patients with nasopharyngeal carcinoma in terms of relapses / secondarisms. It would be very interesting scientifically to compare the variables under study with the Overall Survival and La Desease Free survival. These indices offer more clinically useful information. If this is not possible a brief explanation in the discussions should be made 

Author Response

Thanks you for the insightful comments. We responded to each comment and listed the corresponding responses in the attachment.
